# A Case of Obsessive-Compulsive Disorder Triggered by the Pandemic

**Ana Costa \*, Sabrina Jesus, Luís Simões, Mónica Almeida and João Alcafache**

Department of Psychiatry and Mental Health, Baixo Vouga Hospital Centre, 3810-193 Aveiro, Portugal; 71848@chbv.min-saude.pt (S.J.); 71872@chbv.min-saude.pt (L.S.); 70027@chbv.min-saude.pt (M.A.); 11488@chbv.min-saude.pt (J.A.)

\* Correspondence: 71525@chbv.min-saude.pt or rodrigues.anacosta@gmail.com; Tel.: +351-910-827-825

**Abstract:** Background: The pandemic caused by the sars-cov2 coronavirus can be considered the biggest international public health crisis. Outbreaks of emerging diseases can trigger fear reactions. Strict adherence to the strategies can cause harmful consequences, particularly for people with pathology on the spectrum of obsessive-compulsive disorder. Case presentation: We describe the clinical case of a woman, with a history of anxiety disorder, who develops obsessive-compulsive symptoms, she started cognitive behavioral therapy and pharmacological therapy, with appropriate follow-up. Conclusions: The intense focus on the risk of contamination and the adoption of new hygienic behaviors can be internalized as normative and become an enhancing trigger for obsessive thinking and compulsive behaviors. It is an important focus on prevention, early intervention and adequate follow-up, through measures to promote mental health.

**Keywords:** cognitive behavioral therapies; obsessive-compulsive disorder; adjustment disorders; anxiety disorders; comorbidity

## 1. Introduction

The pandemic caused by the SARS-CoV2 (severe acute respiratory syndrome coronavirus 2) coronavirus can be considered the greatest international public health crisis of the modern era. This highly infectious and novel virus gained global attention in late December 2019, when new cases of SARS reported in Wuhan, China. The World Health Organisation (WHO) declared a public health emergency of international concern on 30 January 2020. As of January 2021, there have been over 93,000,000 cases of coronavirus reported to the WHO, including over 2,000,000 deaths [1,2].

The disease caused by a coronavirus, COVID-19 (coronavirus disease 2019), can cause a severe form of acute respiratory syndrome and quickly lead to death in those vulnerable. As of March 2020, approximately 136 countries have imposed measures to prevent the spread of the virus, including restrictions on free movement with confinement in the home and physical distancing. These measures were accompanied by extensive public health campaigns, targeting, among others, topics such as regular hand washing, hygiene and use of personal protection material [3–5].

Outbreaks of emerging diseases, such as COVID-19, can provoke important reactions of fear, anxiety and worry in the general population. Fear is a conscious subjective experience that involves idiosyncratic concerns and fluctuations over time, being influenced by psychological, sociological and genetic factors. It is an adaptive emotion that serves to mobilize energy to deal with a potential threat. This experience usually triggers security behaviors, such as hand washing, as a way to mitigate certain threats, in this case, fear of contamination [6].

When fear is combined with excessive worry, it typically manifests through a cascade of negative and potentially catastrophic thoughts. Concerns, common among people exposed to an outbreak of infectious disease, often become excessive, especially when

there is physical proximity to individuals categorized as a potential risk group, leading to the adoption of protective measures that are often maladaptive and often associated with comorbidities such as anxiety disorders or depressive mood [6].

During this pandemic, the rapid and global spread of information coupled with the mandatory quarantines, rising mortality numbers and the media's constant focus on the negative impact on communities may have led to excessive/disproportionate social concerns. These can arise from either an excess of information or a fear of the unknown and an uncertain and ongoing threat can become chronic and costly. Additionally, in previous disease outbreaks (e.g., H5N1 avian flu and Zika virus), excessive exposure to the media has been linked to increased fear and, consequently, to all associated effects [6–9].

Increased levels of stress and anxiety since the onset of the pandemic have conditioned, for example, in addition to other various psychological disorders, a greater risk of substance abuse. It was also possible to observe that people previously affected by anxiety-related disorders are more affected by stressors related to the pandemic, compared to those without any diagnosed mental health disorder [4,5,10].

In addition, the implemented essential protection measures can have negative consequences on the mental health of the population. Strict adherence to adopted public health strategies can have harmful consequences, particularly for people with pathology on the spectrum of obsessive-compulsive disorder (OCD). When affected by the fear of contamination, rituals associated with cleaning and disinfection are often created, which are directly related to adaptive behaviors that were observed during the COVID-19 pandemic. Likewise, social distancing, seen as an essential component of infection control, mimics the avoidance behaviors recorded in almost 80% of patients with contamination obsessions. Anecdotally, some authors have reported increased numbers of patients with obsessive-compulsive disorder (OCD) or personality difficulties seeking psychiatric help in recent months. Predictably, their conditions may be exacerbated by the fear of contagion and of loved ones falling ill or feelings of emptiness when quarantined by others [2,5,10–12].

Thus, at least some characteristic symptoms of OCD, more specifically cleaning and disinfection rituals, have evolutionary parallels with behaviors that confer a survival advantage in infectious disease outbreaks, demonstrating a close link between some dimensions of OCD and certain behaviors that evolved to protect humanity from infectious diseases [11,13].

OCD, according to the Diagnostic and Statistical Manual of Mental Disorders, fifth edition (DSM-5), is characterized by the presence of obsessions, compulsions or both [14].

Obsessions consist of recurrent and persistent thoughts, impulses, or images that, at some point during the disturbance, are experienced as intrusive and unwanted and which, in most individuals, cause marked anxiety and suffering. Usually, the individual tries to ignore or block such thoughts, impulses or images or neutralize them with another thought or action, the compulsions [15].

Compulsions are defined by repetitive behaviors or mental acts that the individual feels compelled to perform in response to an obsession. These behaviors or mental acts aim to prevent or reduce anxiety, suffering or avoid a feared event or situation, however, these behaviors or mental acts do not have a realistic connection with what they aim to neutralize or avoid or are excessive [15].

The Yale-Brown Obsessive-Compulsive Scale (Y-BOCS) is a semi-structured interview designed to measure symptom characteristics and severity in OCD patients. It entails 10 items measuring the severity of obsessions and compulsions (Y-BOCS Severity). YBOCS Severity items are scored on a five-point scale from 0 (no symptoms) to 4 (severe symptomatology). High ratings indicate greater severity of symptoms, with a total score from 0 to 40 points, with obsessions and compulsions scoring from 0 to 20 points [14].

The most frequent themes of compulsions are cleanliness, repetition, verification, order and symmetry or mental compulsions, such as praying silently or repeating sentences. As for the etiology, this is associated with concurrent genetic and environmental factors [3,5,15].

While the lifetime prevalence of obsessive-compulsive symptoms is greater than 25%, the lifetime prevalence of the disorder is much lower, estimated at 2–3% for the general population. The rate is somewhat higher in females than in males in adulthood, although the latter is most commonly affected in childhood. The average age of onset of this disorder is 19.5 years and 25% of cases start before 14 years of age, with onset after 35 years being uncommon. Males have an earlier age of onset than females: around 25% of men have the disorder before 10 years of age. The onset of symptoms is usually gradual; however, acute onset has also been reported [15].

OCD is associated with a reduced quality of life and is often co-morbid with anxiety and mood (affective) disorders, namely depressive disorder and is associated with significant impairment in functioning. The WHO ranked OCD within the top ten disabling disorders is associated with dysfunction and decreased quality of life [3,5].

## 2. Case Presentation

In this article, we describe the clinical case of a 52-year-old Caucasian woman. Divorced, she lives with her current partner and works as an administrative assistant in a hospital institution.

The patient in question is followed by psychiatric consultation since adolescence for anxiety disorder, diagnosed according to DSM-5 criteria, translated into excessive anxiety and worry related to various events or daily activities, without the need for medication to control symptoms, by maintaining relative clinical stability. It should be noted as relevant personal history, follicular mucinosis, which is why the patient is medicated with hydroxychloroquine. She does not present risk factors or vulnerability for infection by the SARS-CoV-2 virus [16].

In March 2020, the patient, initially observed by Occupational Medicine, is referred to a psychiatric consultation due to presenting a depressive mood related to frequent and disabling anxiety attacks, which she describes as a tightness in the chest, difficulty in breathing that blocks her and that she associates with "fear of the virus", "much fear of contamination and of contaminating", referring concerns mainly directed to her partner, because of his lung disease and her elderly parents.

At assessment, it is possible to identify recurrent intrusive thoughts of contamination, which are associated with significant levels of anxiety. Additionally, she presents with compulsive washing/cleaning behaviors (e.g., washing of hands and surfaces that may be contaminated—door handles, drawer handles, etc.), excessive care with the demarcation of dirty/clean circuits within the home, disinfection of clothes and any items from the exterior, occupying more than 1 h daily in these rituals.

There are also avoidant and safety and reassurance behaviors, including asking the partner to handle food when shopping at the supermarket, despite recognizing that he is at risk, as well as performing the disinfection of all products before they enter the home.

In her professional sphere, she was unable to finish tasks due to the need for constant hand washing, disinfection of objects and due to feeling frankly distressed taking into consideration the hospital environment where she works ("it seems that wherever I looked I saw the virus hovering"), setting great impact on your professional functionality. At home, the rules and functioning imposed also led to a significant change in family dynamics, impacting a significant loss.

The fact that the patient works in a hospital may have intensified the symptoms as she would consider it to be a source of contagion due to the presence of COVID-19 patients.

During this period, the patient maintained a critical judgment regarding her behavior, realizing that they would be excessive and disproportionate to the risk, even paradoxical, since her partner possesses vulnerability factors for the disease and he would be the one manipulating the objects during shopping.

Physical examination showed extensive erythematous-scaly skin lesions on the upper limbs, especially of the hands and forearms, characteristics of contact eczema secondary to excessive washing and disinfection.

The patient does not have any urinary/bladder or intestinal symptoms.

No analytical changes, namely of the renal, liver, or thyroid function.

According to the case presented and according to the DSM-5, she was diagnosed with OCD and scored 30, in a total of 40 points on the Y-BOCS scale.

The patient began follow-up in cognitive-behavioral therapy, with particular emphasis being placed on Acceptance and Commitment Therapy. Initially with weekly frequency, extending the interval posteriorly to a fortnightly plan and a subsequent monthly session. This process occurred concurrently with psychopharmacological treatment with a selective serotonin reuptake inhibitor in increasing dose, escitalopram 10 mg/day which increased to 40 mg/day for 5 months, complemented with an anxiolytic drug, mexazolam 1 mg at bedtime, with a certificate of temporary incapacity for work issued. In mid-August, there was a significant improvement in mood and reduction of anxiety symptoms, while maintaining obsessive-compulsive rituals of contamination/cleansing, preserving criticism for them, perceiving them as excessive. During cognitive-behavioral psychotherapy sessions, gradual exposure to situations of potential stress was promoted, such as handling objects, with response prevention, avoidance of handwashing always sought to be done in activating environments, such as in-person consultations at the health institution and shopping trips. Additionally, the reduction of ritualistic behaviors such as hand washing. In addition, techniques to decenter their intrusive thoughts were used and their tolerance to uncomfortable physical emotions and sensations was increased through the use of mindfulness practice.

Later, at a time that coincided with the global worsening of the pandemic and the death of a family member from pneumonia caused by SARS-CoV-2, there was an aggravation of obsessive-compulsive symptoms and relapse of depressive symptoms, which intensified the frequency of psychotherapeutic sessions and was associated with the introduction of a tricyclic antidepressant, clomipramine 25 mg, recognized for its anti-obsessive effect, with a favorable response [17].

## 3. Discussion

The global emergence of COVID-19 may be conditioning the aggravation of symptoms in people previously diagnosed with OCD, in the same way, that it may be leading previously healthy individuals to develop characteristic symptoms of this entity, given that these symptoms may be precipitated by stressful life events. The intense focus on the risk of contamination, with the consequent disruption of personal health and social routines, may be associated with triggering obsessive-compulsive symptoms [18].

The adoption of new hygienic behaviors, such as cleaning groceries, produces changes in normal routines. These necessary changes are affecting activities of daily living; however, if integrated as normative and maintained beyond the pandemic may have negative effects on mental health. The normalization of ritualistic hygienic precautions can become a potentiating trigger for obsessive thinking and compulsive behaviors [18].

The fear of illness or of coming into contact with sick individuals often results in high levels of anguish and anxiety, which in turn induce ritualistic behaviors. These behaviors can momentarily reduce anxiety and anguish, translating into a perpetuating cycle (obsession/compulsion) that is often motivated by the overestimation of threats [18].

It is also important to consider the possibility that the supposed obsessive-compulsive symptoms in the pandemic context are an adaptive response for the patient to protect himself and others from the virus, as these behaviors are following public health recommendations. To assess the adaptive/maladaptive character of OCD symptoms during the COVID-19 pandemic, the persistence or resolution of the same symptoms should be assessed in the pandemic recovery phase.

Exposure with response prevention is considered one of the most effective treatment modalities for OCD, as it involves the individual's gradual exposure to the source of fear or anxiety. This intervention may not be feasible due to the nature of COVID-19, as there is a real need to reduce the risk of contracting the virus and to implement restrictions such as

social distancing. During these periods, it can be nearly impossible for people with OCD to confront their fears.

In the present case, given the symptoms, some diagnostic hypotheses were considered. The possibility of anxiety disorder was considered since the symptoms presented (recurrent thoughts, avoidances and constant need for reassurance) coincide with this hypothesis. However, these recurrent thoughts are irrational and are associated with compulsions. The diagnosis of specific phobia was also considered, however, in this case, the fear reaction to specific objects or situations is much more limited and rituals are not present. It is essential to make a differential diagnosis with major depressive disorder, since, along with depressive mood, recurrent/ruminative thoughts are identified, however, in depressive disorder, thoughts are generally congruent with mood and not necessarily experienced as intrusive, in addition to these ruminations not being associated with compulsions, as is typical in OCD.

Unlike the history of anxiety disorder, neither follicular mucinosis nor hydroxychloroquine is a vulnerability factor for the development or worsening of OCD.

However, it should not be assumed that people's vulnerability to OCD is increasing due to washing and disinfecting behaviors observed during the pandemic, as it is their intensity and dysfunctionality that determines their adaptive or maladaptive character.

Every day, for months, we have been bombarded with news that stimulates fear and encourages us to adopt safety/preventive behaviors, namely, washing and disinfecting hands and objects, avoiding human contact through social distancing. All these measures lead us to question which is the ideal level of fear, given that risk is real, but whose preventive measures (if taken excessively) lead to another risk of developing or worsening psychiatric illnesses, such as anxiety, depression, or obsessive-compulsive disorders.

With many people infected but asymptomatic, it makes it impossible to know effectively and immediately whether a person you are in contact with is infected or not, adding more uncertainty to the situation. Intolerance to uncertainty arises when the unknown is intensely perceived resulting in anxiety. Fear of the unknown appears to be a fundamental fear and is a central component of anxiety. Fears related to COVID-19 refer not only to ignorance about the virus but also to the anxiety that accompanies situations that are unpredictable and uncontrollable. Therefore, the fear of this undetectable threat is easily internalized, regardless of the probability of its occurrence.

Another issue to reflect upon is the treatment of this pathology under these circumstances. The literature describes cognitive behavioral therapy with exposure and response prevention as first-line treatment, however, to what extent can we use exposure to real danger as a treatment? Is it ethical/lawful to submit patients to treatment to the detriment of what is advised to the general population as prevention?

## 4. Conclusions

Although few data are describing the prevalence of obsessive-compulsive symptoms during communicable disease pandemics, it is indisputable that they represent periods in which the population is in a state of hypervigilance regarding the prevention of the threat of contamination of itself and others.

The increase in anxiety associated with specific factors and concerns with the theme of contamination can result in cognitive ruminations of a negative connotation, as well as the normalization of behaviors, previously considered ritualistic/compulsive.

The standardization of these new procedures should lead mental health professionals to pay special attention to the assessment of the following questions: are these practices being carried out to neutralize the risk of infection or to reduce obsessive thoughts and anxiety? What behaviors are practiced for effective safety and which are just compulsions?

The elevated levels of anxiety and depression symptoms that have been evidenced highlight the need to focus on prevention, early intervention and adequate follow-up, through measures, to promote the mental health of vulnerable groups most affected during the COVID-19 pandemic.

**Author Contributions:** Conceptualization, A.C., S.J. and M.A.; methodology, A.C., M.A. and L.S.; software, A.C. and S.J.; validation, J.A., M.A. and L.S.; formal analysis, M.A. and J.A.; investigation, A.C.; resources, A.C.; data curation, A.C.; writing—original draft preparation, A.C.; writing—review and editing, S.J.; visualization, M.A.; supervision, M.A.; project administration, J.A.; All authors have read and agreed to the published version of the manuscript.

**Funding:** This research received no specific grant from any funding agency, commercial or not-for-profit sectors.

**Institutional Review Board Statement:** The study was conducted according to the guidelines of the Declaration of Helsinki, and approved by the Institutional Ethics Committee of Baixo Vouga Hospital Centre (protocol code 05-04-2021 and approval at 26 November 2021).

**Informed Consent Statement:** Informed consent was obtained from all subjects involved in the study.

**Data Availability Statement:** No new data were created or analyzed in this study. Data sharing is not applicable to this article.

**Conflicts of Interest:** The authors declare that they have no competing interests.

## Abbreviations

| | |
|---|---|
| COVID-19 | Coronavirus disease 2019 |
| DSM-5 | Diagnostic and Statistical Manual of Mental Disorders, fifth edition |
| OCD | Obsessive-compulsive disorder |
| SARS-CoV2 | Severe acute respiratory syndrome coronavirus 2 |
| SARS | Severe Acute Respiratory Syndrome |
| WHO | World Health Organisation |
| Y-BOCS | Yale-Brown Obsessive-Compulsive Scale |

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
