# Peer review of "A Case of Obsessive-Compulsive Disorder Triggered by the Pandemic"

_psych, doi:10.3390/psych3040055_

Round 1

Reviewer 1 Report

The authors discuss the case of a female patient diagnosed with COVID-19 within the context of the COVID-19 pandemic. The manuscript focuses on an important topic: the risks the COVID-19 pandemic poses on mental health. However, clarifications and modifications are needed. Moreover, the language quality should be improved.
1.   In the medical history, you have described the psychiatric symptoms the patient was consulted for prior to the OCD diagnosis. Please state not only the symptoms but also the diagnosis that the patient was treated for before the onset of OCD and what the treatment entailed.
2.    Please indicate which diagnostic instruments were used to make the OCD diagnosis in 2020.

3.  In the introduction you refer to the situation with the COVID-19 pandemic as of March 2020. Please update this information with the time of manuscript resubmission.

4. Please replace reference 12 with the official DSM-5 reference from 2013 and check for volume and page updates on references from 2020.

5. In my opinion, the quality of the English language should be improved for better flow and readability.

Author Response

  1.  In the medical history, you have described the psychiatric symptoms the patient was consulted for prior to the OCD diagnosis. Please state not only the symptoms but also the diagnosis that the patient was treated for before the onset of OCD and what the treatment entailed.

Anxiety disorder, diagnosed according to DSM-5 criteria

  1. Please indicate which diagnostic instruments were used to make the OCD diagnosis in 2020.

According to the case presented and according to the DSM-5, he was diagnosed with OCD.

  1.  In the introduction you refer to the situation with the COVID-19 pandemic as of March 2020. Please update this information with the time of manuscript resubmission.

The World Health Organisation (WHO) declared a public health emergency of international concern on January 30, 2020. As of January 2021, there have been over 93 000 000 cases of coronavirus reported to the WHO, including over 2 000 000 deaths. 

  1. Please replace reference 12 with the official DSM-5 reference from 2013 and check for volume and page updates on references from 2020.

American Psychiatric Association. Diagnostic and Statistical Manual of Mental Disorders Fifth Edition. American Psychiatric Publishing. 2013.

  1. In my opinion, the quality of the English language should be improved for better flow and readability.

Adjusted

Reviewer 2 Report

This case report outlines a possible clinical effect related to the ongoing COVID-19 pandemic. Please see below for my specific comments.

Specific comments:

  1. The title of the paper should be "A Case of Obsessive-Compulsive Disorder Triggered by the Pandemic".
  2. Please define the abbreviations "COVID-19" and "SARS-CoV-2" in the first instance of their use. Note that "COVID-19" is actually short for "Coronavirus Disease 2019".
  3. Anecdotally, some authors have reported increased numbers of patients with obsessive-compulsive disorder (OCD) or personality difficulties seeking psychiatric help in the recent months. Predictably, their conditions may be exacerbated by the fear of contagion and of loved ones falling ill or feelings of emptiness when quarantined from others (citation: pubmed.ncbi.nlm.nih.gov/34171975 and pubmed.ncbi.nlm.nih.gov/32380875). These should be mentioned in the introduction.
  4. Please change "she started follow-up on cognitive-behavioural therapy along with psychopharmacological" to "she started cognitive behavioural therapy and pharmacological therapy, with appropriate follow up."
  5. How was her functioning at home and in the workplace?
  6. How long have the symptoms been going on for? This was not clearly specified.
  7. Did the patient exhibit any urinary/bladder or bowel symptoms? There is also a proposed subtype of OCD patients who have urinary obsessions and they may report predominant fear of urinary incontinence and urinary frequency without organic etiology (citation: ncbi.nlm.nih.gov/pmc/articles/PMC8235037).
  8. Could this patient experience even more fear of contamination and repeated reminders of strict hygienic behaviours given her workplace (hospital)?
  9. Please present the relevant laboratory results (and reference ranges) in a table format.
  10. "... reports and calculations of the lethality rate are impossible to be carried out accurately" - this is not true (and also not really relevant to the case discussion). In fact, this means that the lethality rate is lower than expected since there exists a group of asymptomatic individuals who are not tested/diagnosed with COVID-19.

Author Response

1- The title of the paper should be "A Case of Obsessive-Compulsive Disorder Triggered by the Pandemic": Adjusted

2- Please define the abbreviations "COVID-19" and "SARS-CoV-2" in the first instance of their use. Note that "COVID-19" is actually short for "Coronavirus Disease 2019": Adjusted

  • COVID-19: Coronavirus disease 2019
  • SARS-CoV2: Severe acute respiratory syndrome coronavirus 2

3- Anecdotally, some authors have reported increased numbers of patients with obsessive-compulsive disorder (OCD) or personality difficulties seeking psychiatric help in the recent months. Predictably, their conditions may be exacerbated by the fear of contagion and of loved ones falling ill or feelings of emptiness when quarantined from others (citation: pubmed.ncbi.nlm.nih.gov/34171975 and pubmed.ncbi.nlm.nih.gov/32380875). These should be mentioned in the introduction: Adjusted

4- Please change "she started follow-up on cognitive-behavioural therapy along with psychopharmacological" to "she started cognitive behavioural therapy and pharmacological therapy, with appropriate follow up.": Adjusted

5- How was her functioning at home and in the workplace?: In her professional sphere, she was unable to finish tasks due to the need for constant hand washing, disinfection of objects and due to feeling frankly distressed taking into consideration the hospital environment where she works (“it seems that wherever I looked I saw the virus hovering”), setting great impact on your professional functionality. At home, the rules and functioning imposed also led to a significant change in family dynamics, impacting a significant loss.

6- How long have the symptoms been going on for? This was not clearly specified: He describes the onset of symptoms in mid-January 2020 and in March, when he sees the doctor, he has several functional limitations in his workplace.

7- Did the patient exhibit any urinary/bladder or bowel symptoms? There is also a proposed subtype of OCD patients who have urinary obsessions and they may report predominant fear of urinary incontinence and urinary frequency without organic etiology (citation: ncbi.nlm.nih.gov/pmc/articles/PMC8235037): the patient does not have any urinary/bladder or intestinal symptoms.

8- Could this patient experience even more fear of contamination and repeated reminders of strict hygienic behaviours given her workplace (hospital)?.: The fact that the patient works in a hospital may have intensified the symptoms as she would consider it to be a source of contagion due to the presence of covid-19 patients.

9- Please present the relevant laboratory results (and reference ranges) in a table format: not applicable

10- "... reports and calculations of the lethality rate are impossible to be carried out accurately" - this is not true (and also not really relevant to the case discussion). In fact, this means that the lethality rate is lower than expected since there exists a group of asymptomatic individuals who are not tested/diagnosed with COVID-19: With many people infected but asymptomatic, it makes it impossible to know effectively and immediately whether a person you are in contact with is infected or not, adding more uncertainty to the situation.

Reviewer 3 Report

In the present paper, the Authors described the clinical case of a woman, with a history of anxiety disorder, who developed obsessive-compulsive symptoms triggered by the pandemics, and started follow-up on cognitive-behavioural therapy along with psychopharmacological interventions. 

Overall, I found the present paper very interesting and scientifically sound. I have only some minor comments aimed to improve the quality of the manuscript and these are outlined below:

1) I believe that in the introduction a brief note on possible anxiety disorders and PTSD developed by healthcare workers should be added due to the pandemic (as the patient was an administrative assistant in a hospital institution, not merely an HCW but similar). Please, refer to doi 10.9758/cpn.2021.19.4.780.

2) A comorbid adjustment disorder diagnosis would also fit well with the clinical picture of the patient in my opinion. What the Authors think about this?

3) I believe that one of the mechanisms that might be involved in triggering OCD and ruminations is the increased salience toward the virus and the possibility of contagion. This increased salience might be associated also with OCD with poor or absent insight or, even, psychosis. Please add a brief consideration on this point with appropriate references (see dois 10.1016/j.euroneuro.2021.10.004 and 10.3390/jcm10153441).

4) Have the Authors administered any rating scales the can objectively measure and give us a detailed score on the patient' distress?

Author Response

1- Healthcare workers are in fact exposed to different risks from the general population, being more susceptible to the development of symptoms related to post-traumatic stress, however, this entity is not the scope of this article.

2-  Although the patient presents symptoms compatible with adjustment disorder, namely, emotional and behavioral symptoms in response to an easily identifiable stressor, however, along with this symptomatology, she also presents obsessions followed by compulsions that impact her family, professional and social life that are features of obsessive-compulsive disorder.

3- The patient maintained her critical judgment throughout the process and never developed symptoms of the psychotic line

4- Yes, The Yale-Brown Obsessive-Compulsive Scale (YBOCS) questionnaire was applied, which is referenced in the manuscript

Round 2

Reviewer 1 Report

Thank you for addressing my comments and for the revised version of the manuscript. You have answered all factual comments/questions. However, in my opinion, the quality of the language should be further improved. There are many instances of language inconsistencies. For example, in line 155 you refer to the patient as "he", while you have stated that she was a female; in lines 146/147 there is an inconsistency in the use of the plural and singular form; and in line 35 the serial comma has not been used. I would recommend editing the text professionally to ensure that the quality of the language is sufficient for publication.

Author Response

Thank you for your appreciations and comments. As suggested i will edit the text professionally to fix theses flaws

Reviewer 2 Report

Thank you for the revisions.

Specific comments:

  1. The manuscript is in need of wordsmithing still. 
  2. "... even paradoxical, since her partner possesses vulnerability factors for the disease, and he would the one manipulating the objects during shopping" - the phrasing here is very convoluted and awkward. Please rephrase.
  3. "... he was diagnosed with POC" - what does 'POC' stand for?
  4. "... he was diagnosed with" - the patient is a 'she' right?
  5. Major depressive disorder is the most frequent comorbid condition in obsessive-compulsive disorder. This should be mentioned.

Author Response

Thank you for your appreciations and comments. As suggested i will edit the text to fix theses flaws.